# Comparative Clinical Outcomes of Major Respiratory Viruses in Hospitalized Adults During the Post-Pandemic Period: A Retrospective Cohort Study

**DOI:** 10.3390/v17121545

**Published:** 2025-11-26

**Authors:** Hasip Kahraman, Gizem Keser, Furkan Süha Ölmezoğlu, Betül Altıntaş Öner, Onur Sedat Kurt, Tercan Us, Fatma Erdem

**Affiliations:** 1Department of Infectious Diseases and Clinical Microbiology, Eskişehir Osmangazi University Faculty of Medicine, Eskisehir 26040, Türkiye; drgzmksr@gmail.com (G.K.); suhafurkan23@hotmail.com (F.S.Ö.); onursedatk@gmail.com (O.S.K.); 2Department of Infectious Diseases and Clinical Microbiology, Eskişehir City Hospital, Eskişehir 26080, Türkiye; betulaltintas4390@gmail.com; 3Department of Clinical Microbiology, Eskişehir Osmangazi University Faculty of Medicine, Eskişehir 26040, Türkiye; tercanus57@gmail.com (T.U.); fatma.erdem@ogu.edu.tr (F.E.)

**Keywords:** respiratory tract infections, influenza, SARS-CoV-2, RSV, multiplex PCR

## Abstract

Background: In the post-pandemic era, respiratory viruses continue to cause substantial morbidity and mortality among hospitalized adults. SARS-CoV-2 and influenza remain the most common pathogens, while RSV and rhinovirus have re-emerged as relevant causes of severe illness. This study compared the characteristics and outcomes of virus-specific infections detected by multiplex real-time PCR over two consecutive seasons. Methods: This retrospective cohort study was conducted at a 1010-bed tertiary-care hospital in Türkiye between June 2022 and June 2024. Adults hospitalized with at least one respiratory virus detected by MRT-PCR were included. Demographic, clinical, and laboratory data were analyzed. Pathogen-specific comparisons were limited to monoinfections, and predictors of in-hospital mortality were identified using multivariable logistic regression. Results: Among 518 admissions, influenza (33.6%) and SARS-CoV-2 (29.3%) were the predominant pathogens, followed by rhinovirus (11.2%), RSV (6.6%), and other respiratory viruses (19.6%). Overall in-hospital mortality was 26.6%. Mortality differed across virus groups in unadjusted analyses, being highest in SARS-CoV-2 and RSV and lowest in rhinovirus. Non-survivors were older, more comorbid, more often immunosuppressed, and more likely to require oxygen therapy or ICU care at sampling. In multivariable analysis, independent predictors of mortality were ICU location at sampling (aOR 5.52), oxygen requirement (aOR 3.39), immunosuppression (aOR 3.67), older age (per 10-year increase: aOR 1.25), and secondary bacterial infection (aOR 7.00). Viral etiology, including SARS-CoV-2, was not independently associated with mortality after adjustment. Conclusions: Among hospitalized adults, mortality was driven primarily by host-related factors and secondary bacterial infection rather than by viral etiology. These findings highlight the need for strengthened adult immunization programs, reliable respiratory virus surveillance, the prevention of bacterial superinfection, and the development of and equitable access to effective vaccines and antiviral therapies to reduce severe outcomes in high-risk adults.

## 1. Introduction

Respiratory viral infections are a major driver of acute respiratory illness worldwide and remain a leading cause of lower respiratory infection morbidity and mortality, particularly among adults requiring hospitalization [1]. Viral pathogens—including influenza viruses, coronaviruses (including SARS-CoV-2), respiratory syncytial virus (RSV), and Human rhinoviruses (HRV)—are among the most frequently identified etiologies in inpatient settings [2]. In hospitalized adults with acute respiratory illness, clinical severity ranges from ward care to intensive care, underscoring challenges in early etiologic attribution and risk stratification [3].

Among respiratory viruses, influenza remains one of the most clinically significant pathogens, particularly types A and B, which are responsible for recurrent seasonal epidemics worldwide. The clinical spectrum of influenza infection ranges from mild upper respiratory tract disease to severe viral pneumonia with fatal complications. According to the World Health Organization (WHO), seasonal influenza causes approximately one billion cases annually, including three to five million severe illnesses and 290,000–650,000 deaths [4].

Severe acute respiratory syndrome coronavirus 2 (SARS-CoV-2) has had a profound global impact, with more than seven million deaths reported to the WHO, and it has reshaped the epidemiology of viral respiratory infections [5]. The clinical spectrum of COVID-19 ranges from mild upper respiratory tract symptoms to severe viral pneumonia and acute respiratory distress syndrome, often necessitating hospitalization and intensive care support [6]. Although influenza and SARS-CoV-2 share overlapping clinical features, comparative data on their respective impact in adult patients remain relatively limited.

RSV is increasingly recognized as an important cause of respiratory infections in adults, particularly among older adults and immunocompromised individuals, where it is associated with a higher risk of severe disease and hospitalization [7]. HRV also contributes substantially to the overall burden of respiratory tract infections and account for a wide clinical spectrum, ranging from common colds to more severe lower respiratory tract involvement. The clinical manifestations of HRV are often nonspecific, making it difficult to distinguish from other viral respiratory infections [8]. In inpatient settings, RSV is typically detected less often than influenza and SARS-CoV-2 among adults, whereas HRV detection varies widely by season and setting; nevertheless, both viruses remain clinically relevant given their impact on older and immunocompromised patients [9,10].

Multiplex real-time PCR (MRT-PCR) is a widely adopted diagnostic approach that offers rapid turnaround, high analytical sensitivity and specificity, and simultaneous detection of multiple respiratory viruses from a single specimen. This approach enables the identification of both common pathogens, such as influenza and coronaviruses, and less frequently detected agents, including adenoviruses, human metapneumovirus, and parainfluenza viruses, which are often missed by conventional techniques due to lower sensitivity or prolonged turnaround [11,12,13].

We aimed to describe the distribution of respiratory viruses in hospitalized adults using MRT-PCR of upper respiratory tract specimens, compare clinical/laboratory features across etiologies, and assess their association with major in-hospital outcomes.

## 2. Methods

### 2.1. Study Design and Setting

This retrospective cohort study was conducted at Eskişehir Osmangazi University Faculty of Medicine (Eskişehir, Türkiye), a 1010-bed tertiary-care referral hospital with multiple intensive care units, between 1 July 2022 and 30 June 2024.

### 2.2. Patient Population

All hospitalized adults who underwent MRT-PCR testing of an upper respiratory tract specimen during the study period were screened. Patients were included if ≥1 respiratory virus was detected and excluded if only panel bacterial targets were positive (e.g., *Legionella* spp., *Bordetella pertussis*) or if key clinical/outcome data were missing. When multiple positive MRT-PCR results occurred within the same admission, only the first (“index”) result was retained; the unit of analysis was the admission. In total, 518 virus-positive admissions were analyzed, as summarized in Figure 1. No formal sample size calculation was performed due to the retrospective nature of the study.

### 2.3. Data Collection

Demographic characteristics (age, sex), comorbidities, and vaccination status were retrieved from electronic medical records. Comorbidity burden was summarized using the Charlson Comorbidity Index (CCI), calculated according to the original scoring system based on comorbidities present at admission. Clinical data, including presenting symptoms, vital signs, ward/ICU status, and respiratory support, were abstracted at prespecified time points: Day 0 (index ±24 h), Day 5–7, Day 14, and Day 28. Key laboratory findings—white blood cell count (WBC), neutrophil count, absolute lymphocyte count (ALC), and C-reactive protein (CRP)—were collected only at Day 0 and Day 5–7. Chest imaging was obtained at the treating clinician’s discretion. Radiological pneumonia was captured from radiology reports or clinical documentation when available; missing values were not imputed. Treatment modalities and clinical outcomes (secondary bacterial infections, length of hospital stay, ICU admission, need for mechanical ventilation, and in-hospital mortality) were recorded. For descriptive temporal summaries, detections were aggregated according to the specimen-collection date. All data were manually entered into a standardized case report form by two investigators and double-checked for accuracy; datasets were de-identified before analysis.

### 2.4. Definitions

Vaccination status was operationalized relative to the sampling date; all vaccinations had to be completed ≥1 month before the index date. Specifically: influenza ≥1 dose in the prior 12 months; COVID-19 ≥2 mRNA doses or ≥3 inactivated doses; pneumococcal ≥1 dose PCV13. Omicron-adapted (bivalent) COVID-19 vaccines were not available in Türkiye during the study period. Immunosuppression was defined as active cancer therapy or transplant; systemic corticosteroids ≥10 mg prednisolone-equivalent for ≥14 days; cytotoxic or biologic immunomodulators; primary immunodeficiency; or advanced HIV infection. Patients meeting any of these criteria were classified as immunosuppressed. Fever was defined as a measured temperature ≥38.0 °C. Antiviral therapy was defined as receipt of a pathogen-appropriate agent (oseltamivir for influenza; molnupiravir for SARS-CoV-2) during the index hospitalization; early initiation was defined a priori as ≤48 h from the time of specimen collection, as symptom-onset timing could not be reliably determined for all patients. Oxygen support was the maximum level received at any time during the index admission. Mixed infection (co-detection): ≥2 viruses detected in the same sample; monoinfection = 1 virus. Secondary bacterial infection was defined as clinically significant bacterial growth in blood and/or respiratory cultures obtained during hospitalization. In-hospital mortality was defined as all-cause death occurring during the index hospitalization, prior to discharge; post-discharge deaths were not evaluated.

### 2.5. Microbiological Diagnosis

Nasopharyngeal swab specimens were collected by healthcare personnel trained in standard procedures for the safe handling of biological materials and transferred into the transport medium supplied by the kit manufacturer (QIAstat-Dx Respiratory SARS-CoV-2 Panel; QIAGEN, Hilden, Germany). All samples were processed on the QIAstat-Dx Analyzer using the manufacturer’s instructions. A volume of 300 µL was loaded into the main port of the panel cartridge with single-use transfer pipettes. The system performs nucleic acid extraction, amplification, and detection based on real-time PCR technology in a fully automated, closed, and contamination-free environment. Results are provided within approximately one hour and include SARS-CoV-2 as well as 21 additional respiratory pathogens (see Appendix A for complete pathogen list) [14,15]. Testing was performed at the discretion of the treating clinician based on respiratory signs or symptoms. Testing practices remained stable throughout the study period, and MRT-PCR ordering was based on consistent symptom-driven indications, without changes in hospital policy. In the MRT-PCR panel, rhinovirus and enterovirus share a combined detection target (reported as rhinovirus/enterovirus). Therefore, all detections are referred to as rhinovirus throughout the text for consistency. SARS-CoV-2 variant analysis was not performed in this study.

### 2.6. Outcomes

The primary outcome was all-cause in-hospital mortality. Secondary outcomes included total length of hospital stay (admission-to-discharge), ICU admission after sampling among patients initially managed on the ward, the highest level of respiratory support required during hospitalization (no oxygen, low-flow, high-flow/reservoir mask, or mechanical ventilation), and the presence of viral co-infections on MRT-PCR. We also evaluated time-bound mortality at days 7, 14, and 28, as well as time from sampling to death and time from admission to death; ICU and ward length-of-stay were summarized separately.

### 2.7. Ethics Approval and Consent to Participate

The study was conducted in accordance with the principles of the Declaration of Helsinki and approved by the Eskişehir Osmangazi University Non-Interventional Clinical Research Ethics Committee, with reference number 2024-228 (approved on 23 July 2024). Since this study did not involve direct interaction with human participants requiring individual consent, the need for informed consent to participate was waived by the ethics committee in accordance with national regulations.

### 2.8. Statistical Analysis

Continuous variables were summarized as mean ± standard deviation (SD) or median (interquartile range, IQR) based on distribution (Shapiro–Wilk) and categorical variables as counts and percentages. Two-group comparisons (e.g., influenza vs. SARS-CoV-2) used the Mann–Whitney U test for continuous variables and Pearson’s chi-square or Fisher’s exact test for categorical variables. Multi-group comparisons across virus categories (influenza, SARS-CoV-2, rhinovirus, RSV, others) used the Kruskal–Wallis test for continuous variables and Pearson’s chi-square test for categorical variables, with pairwise post hoc comparisons adjusted by the Holm method. Univariable analyses first identified factors associated with in-hospital mortality; variables with *p* < 0.10 and clinically relevant covariates were then entered into a multivariable logistic regression to estimate adjusted odds ratios (ORs) with 95% confidence intervals (CIs). Two-sided *p* < 0.05 was considered statistically significant. Statistical analyses were conducted using SPSS version 22.0 (IBM Corp., Armonk, NY, USA).

## 3. Results

### 3.1. Description of the Cohort

During the study period, a total of 4254 patients underwent multiplex respiratory virus PCR testing. Of these, 2440 patients with negative test results were excluded. An additional 1296 patients were excluded due to age <18 years, outpatient status, missing data, or other exclusion criteria. The final analysis included 518 patients, comprising 479 (92.5%) cases with single pathogen infections and 39 (7.5%) cases with mixed infections (Figure 1).

### 3.2. Baseline Characteristics and Distribution of Detected Respiratory Viruses

Among 518 inpatients, the mean age was 62.8 ± 17.6 years and 282 (54.4%) were male. Nearly all had ≥1 comorbidity (476, 91.9%), most commonly diabetes mellitus (145, 28.0%); median CCI was 5 (IQR 2–7), and 223 (43.1%) were immunosuppressed. Vaccination included influenza (37, 7.1%), COVID-19 (469, 90.5%), and pneumococcal (151, 29.2%). At sampling, cough was present in 279 (53.9%), and fever/dyspnea each in 161 (31.1%); 262 (50.5%) required oxygen. Radiological pneumonia was documented in 109 (21.0%). Secondary bacterial infection was identified in 53 patients (10.2%), including 48 with monoinfections and 5 with mixed viral infections. Most samples were obtained on the ward (361, 69.7%) rather than the ICU (157, 30.3%). Among ward-at-sampling patients, 96/361 (26.6%) required ICU transfer; median total length of stay was 16 days (IQR 9–29), and in-hospital mortality was 163 (31.5%). By day 30, 299 (57.7%) were discharged, 52 (10.0%) remained hospitalized, 44 (8.5%) were in ICU, and 123 (23.7%) had died; median time to death was 14 days from sampling and 23 days from admission (Table 1).

The leading detections were SARS-CoV-2 169 (32.6%) and Influenza A 167 (32.2%), followed by rhinovirus 72 (13.9%), RSV A/B 52 (10.0%), and Influenza B 24 (4.6%). Less frequent agents included parainfluenza viruses, human metapneumovirus, seasonal human coronaviruses (229E, OC43, HKU1, NL63), and adenovirus; mixed infections were identified in 39 patients (7.5%) (Table 2). The temporal distribution of detected viruses is illustrated in Figure 2. Influenza (A + B) and RSV displayed pronounced winter peaks during both study years, whereas SARS-CoV-2 showed similar seasonal surges but remained detectable throughout the year. Rhinovirus circulated year-round with intermittent fluctuations, and detections of other respiratory viruses occurred at low levels with irregular distribution.

### 3.3. Clinical and Laboratory Characteristics by Virus

When compared across virus groups (Table 3), immunosuppression was most frequent in RSV (64.7%) and SARS-CoV-2 (47.4%) (*p* < 0.001). Fever was most common in influenza (34.5%) (*p* = 0.005). The highest median WBC was observed in rhinovirus infections (9.9 ×10^3^/µL) (*p* = 0.010). Secondary bacterial infection rates also varied across pathogens, ranging from 5.2% in influenza to 13.2% in SARS-CoV-2, 12.1% in rhinovirus, and 5.9% in RSV infections (*p* = 0.062). Total length of stay was similar across groups (*p* = 0.158). Although both ICU admission after sampling (highest in SARS-CoV-2: 25.7%) and COVID-19 vaccination status showed a significant overall difference across groups (*p* = 0.016 and *p* = 0.035, respectively), no individual pairwise comparisons remained significant after Holm adjustment. In-hospital mortality differed by pathogen and was highest in SARS-CoV-2 (40.8%) (*p* < 0.001). These comparative analyses were restricted to monoinfections to avoid confounding; mixed infections and other respiratory viruses with small case numbers (adenovirus, parainfluenza viruses, human metapneumovirus, and seasonal coronaviruses [229E, OC43, HKU1, NL63]) were excluded.

### 3.4. Comparison Between Influenza and SARS-CoV-2 Infections

Influenza cases were far more likely to receive antivirals (150/174, 86.2% vs. 50/152, 32.9%; *p* < 0.001). Despite this, SARS-CoV-2 showed greater clinical severity: higher oxygen requirement (64.5% vs. 53.4%; *p* = 0.044), more ICU admission (56.6% vs. 41.4%; *p* = 0.006), and higher in-hospital (40.8% vs. 25.9%; *p* = 0.004) and 28-day mortality (28.3% vs. 16.1%; *p* = 0.008). Day-7 lymphopenia was more pronounced with SARS-CoV-2 (ALC 820 vs. 985 cells/µL; *p* = 0.012). Mechanical ventilation and total length of stay did not differ significantly. Analyses were restricted to monoinfections; mixed infections were excluded (Table 4). Additional Day 0/Day 7 laboratory data are provided in Appendix A.

### 3.5. Determining the Factors Affecting Mortality

A univariable comparison was conducted between survivors (n = 355) and non-survivors (n = 163). Non-survivors were older (median 70 vs. 64 years) and had a higher CCI (6.0 vs. 4.0; both *p* < 0.001). Immunosuppression was more frequent among non-survivors (52.8% vs. 38.6%; *p* = 0.003). At sampling, non-survivors more often required oxygen (81.6% vs. 36.3%; *p* < 0.001) and were in the ICU (59.5% vs. 16.9%; *p* < 0.001). They also had higher rates of radiological pneumonia (36.2% vs. 14.1%; *p* < 0.001), mechanical ventilation (54.6% vs. 6.2%; *p* < 0.001), and—among ward-at-sampling patients—ICU transfer (84.8% vs. 13.6%; *p* < 0.001), as well as a markedly higher frequency of secondary bacterial infection (21.5% vs. 5.1%; *p* < 0.001). WBC and CRP levels were higher, while ALC was lower among non-survivors. As noted above, mortality differed by pathogen (*p* = 0.026), being highest in SARS-CoV-2 (40.8%) and RSV (38.2%), intermediate in influenza (25.9%) and other viruses (31.1%), and lowest in rhinovirus (13.8%). Mixed infections occurred in 7.5% of patients and showed no mortality difference (*p* = 0.247). (Table 5).

### 3.6. Independent Predictors of In-Hospital Mortality

Multivariable logistic regression identified ICU location at sampling (aOR 5.52, 95% CI 3.00–10.16; *p* < 0.001), oxygen requirement at sampling (aOR 3.39, 95% CI 1.86–6.18; *p* < 0.001), immunosuppression (aOR 3.67, 95% CI 1.95–6.89; *p* < 0.001), age (per 10 years) (aOR 1.25, 95% CI 1.00–1.57; *p* = 0.048), and secondary bacterial infection (aOR 7.00, 95% CI 3.13–15.66; *p* < 0.001) as independent predictors of in-hospital mortality. Rhinovirus infection remained inversely associated with mortality compared with influenza (aOR 0.29, 95% CI 0.11–0.78; *p* = 0.014). WBC and ALC remained in the multivariable model but showed minimal effect sizes (aOR 1.04 and 0.99, respectively), and were therefore not independently associated with mortality after adjustment. SARS-CoV-2, RSV, other viruses, sex, CCI, and CRP were also not independently associated (Table 6).

## 4. Discussion

In this single-center cohort spanning two consecutive years in the post-pandemic period, SARS-CoV-2 and influenza A predominated among hospitalized adults, whereas rhinovirus and RSV were less frequent, and other respiratory viruses were detected only sporadically. These findings reflect the evolving epidemiology of viral respiratory infections after the COVID-19 pandemic, with influenza and SARS-CoV-2 representing the predominant causes of hospitalization while RSV and rhinovirus circulated at lower levels. The cohort represented a high-risk case mix, characterized by advanced age, multiple comorbidities, and frequent requirements for oxygen therapy and ICU care. This setting provided an opportunity to compare the clinical and epidemiologic features of major respiratory viruses across consecutive post-pandemic seasons, addressing the limited real-world data on the relative burden and presentation of these pathogens in hospitalized adults.

Global surveillance data show that, following the COVID-19 pandemic, respiratory viruses gradually regained their characteristic circulation patterns. As mitigation measures were lifted, influenza and RSV re-established distinct winter peaks, whereas SARS-CoV-2 shifted toward a more endemic, multiseason circulation [6,16]. Rhinovirus continued its characteristic year-round activity [10]. Our findings mirrored these post-pandemic trends, with influenza (A + B) and RSV showing clear winter peaks, SARS-CoV-2 remaining detectable at lower levels year-round, and rhinovirus circulating continuously with intermittent fluctuations. The alignment of our data with global surveillance patterns suggests that the observed distributions reflect genuine epidemiological dynamics rather than artifacts of seasonal testing intensity. Similar seasonal dynamics were reported in Italy during a comparable post-pandemic period in hospitalized adult cohorts, although pathogen-specific frequencies differed between settings [17]. Data from a large tertiary center in China likewise noted increasing influenza A positivity among older adults, consistent with the observed shift toward greater post-pandemic influenza burden in elderly populations [18]. Collectively, these results illustrate the dynamic coexistence of traditional and emerging respiratory pathogens in the evolving post-COVID landscape.

A key question in respiratory virus research concerns the relative contributions of pathogen virulence and host vulnerability to disease severity. Recent evidence indicates that RSV can cause illness comparable in severity to influenza or COVID-19, particularly among immunocompromised adults—challenging its long-standing perception as a “mild” pathogen [7,9]. Similarly, rhinoviruses, though commonly associated with mild upper respiratory infections, have been increasingly recognized as causes of severe lower respiratory disease in high-risk groups [10]. In our cohort, baseline frailty indicators such as age, comorbidity burden, and CCI scores were largely comparable across virus categories, yet the prevalence of immunosuppression was substantially higher among patients with RSV (64.7%) and SARS-CoV-2 (47.4%) than those with influenza (32.8%). This likely contributed to the higher mortality observed in RSV infections within our cohort. Despite these differences in host immune status, key measures of initial severity—including oxygen requirement and ICU status—did not differ significantly. Although literature suggests bacterial superinfection rates can reach 24% in SARS-CoV-2 cases with significant variations between pathogens, our cohort showed only non-significant numerical increases in SARS-CoV-2 and rhinovirus co-infections. This observation suggests that early clinical manifestations were broadly similar across viral etiologies, although subsequent disease trajectories differed markedly.

Multiple recent studies have confirmed that SARS-CoV-2 continues to cause more severe disease than seasonal influenza in hospitalized adults, reflected in higher rates of oxygen requirement, ICU admission, and in-hospital mortality [19,20]. In a nationwide cohort study conducted in the United States during the 2023–2024 winter season, in-hospital mortality remained significantly higher for COVID-19 than for influenza despite similar age and comorbidity profiles [21]. A population-based Danish study spanning 2022–2024 likewise reported longer hospital stays, greater ICU utilization, and approximately twofold higher mortality risk among COVID-19 cases [22]. Consistent with these findings, our cohort also showed markedly higher ICU transfer and mortality rates among COVID-19 cases compared with influenza.

A similar pattern was observed for RSV, which showed disease severity comparable to or exceeding those of influenza and even SARS-CoV-2, especially among older or immunocompromised adults [7,9]. Rhinovirus infections had the lowest mortality rates in our cohort, yet this remained considerable for a pathogen generally regarded as a cause of mild infections. Because the assay target detects rhinovirus and enterovirus jointly, our analysis used the combined category ‘rhinovirus’, and this heterogeneity should be considered when interpreting the findings. Serial laboratory data further supported this pattern, with a more pronounced decline in lymphocyte counts by Day 7 among SARS-CoV-2 cases, suggesting a sustained systemic inflammatory response compared with influenza. Together, these findings emphasize the continued clinical burden posed by SARS-CoV-2 and highlight that RSV has re-emerged as a major cause of severe respiratory illness among vulnerable adults.

The significantly higher frequency of oseltamivir use among influenza cases likely contributed to their lower mortality compared with COVID-19. However, because treatment timing was defined relative to the sampling date rather than symptom onset, this measure reflects prompt in-hospital management rather than accurately capturing the optimal biological window for antiviral efficacy—particularly for influenza, where the benefit of early therapy is time-dependent. Despite the high coverage of COVID-19 vaccination in our cohort, mortality among SARS-CoV-2 cases remained substantial, consistent with recent reports linking persistent vulnerability to age and frailty rather than vaccine failure. Notably, Omicron-adapted vaccines were not introduced in Türkiye, and none of the patients in this study had received variant-specific boosters. This may partly explain the sustained mortality despite prior immunization.

Advanced age, high comorbidity burden, and immunosuppression consistently emerge as dominant predictors of poor outcomes across cohorts involving SARS-CoV-2, influenza, and RSV [23]. In particular, hypoxemia, ICU admission, and elevated inflammatory markers such as C-reactive protein have been closely associated with mortality [24,25]. The large multicenter analyses from Denmark and the United States highlight that frailty indicators—including age, comorbidity, and early oxygen requirement—substantially contribute to mortality risk across viral etiologies [21,22]. In our cohort, which reflects the high clinical complexity of a tertiary-care referral population, non-survivors were older, more comorbid, and more often immunosuppressed, and a considerable proportion were already in the ICU at the time of sampling or required subsequent ICU transfer, indicating substantial severity at presentation and likely contributing to the overall high mortality rate. Radiological evidence of pneumonia was also strongly associated with mortality in unadjusted analyses, underscoring the contribution of lower respiratory tract involvement to adverse outcomes. Notably, the overall mortality (31.5%) exceeded the proportion of patients with radiologically confirmed pneumonia (21.0%). Moreover, because chest imaging was obtained selectively rather than systematically, some cases of lower respiratory tract involvement may have been under-recognized, a limitation inherent to the retrospective design. Consistent with Shafran et al., who identified secondary bacterial infection as a significant predictor of death [26], our cohort demonstrated a markedly higher mortality rate in patients with secondary bacterial infection compared to those without (21.5% vs. 5.1%). Mortality also varied across virus groups (*p* = 0.026), being highest with SARS-CoV-2 and lowest with rhinovirus, with influenza and other viruses showing intermediate rates. Although host frailty contributed substantially to outcome differences, pathogen-related effects were also evident.

In the multivariable model, ICU location at sampling, oxygen requirement, immunosuppression, older age, and secondary bacterial infection emerged as the strongest independent predictors of in-hospital mortality, underscoring the combined impact of baseline frailty, acute physiological compromise, and bacterial superinfection on clinical outcomes. Secondary bacterial infection demonstrated one of the largest effect sizes in our model, consistent with extensive evidence showing that bacterial superinfection markedly increases mortality in respiratory virus infections [26,27]. Contrary to recent multicenter data showing that SARS-CoV-2 still carries a higher mortality risk than influenza—although the gap has narrowed in contemporary clinical settings [21,28]—we observed only a non-significant numerical trend (aOR 1.74; *p* = 0.073) after adjustment for acute physiological severity. Conversely, the inverse association observed for rhinovirus likely reflects its detection in patients with milder respiratory illness. Conversely, the inverse association observed for rhinovirus likely reflects its detection in patients with milder respiratory illness. Taken together, these findings emphasize that host vulnerability, acute disease severity, and secondary bacterial infection outweigh viral etiology in determining mortality among hospitalized adults with respiratory viral infections.

### Strengths and Limitations

This study has several strengths. It provides a comprehensive comparison of major respiratory viruses among hospitalized adults over two consecutive post-pandemic seasons, using syndromic MRT-PCR testing that enables simultaneous pathogen detection with high analytical accuracy. The dataset reflects real-world clinical practice and includes detailed demographic, clinical, and laboratory parameters, allowing both univariable and multivariable analyses across a broad spectrum of viral etiologies. By restricting pathogen-specific comparisons to monoinfections, the analysis minimized misclassification bias due to co-detection. Several limitations should also be acknowledged. The cohort represents a particularly high-risk inpatient population with advanced age, substantial comorbidity burden, and a high prevalence of immunosuppression; therefore, generalizability to younger or community-managed populations may be limited. As a single-center retrospective study, the findings may not be fully generalizable to other healthcare settings. Pathogen-specific comparisons excluded mixed infections to reduce misclassification; descriptive summaries included both. Additionally, because symptom-onset time could not be reliably ascertained for all cases, the “time to treatment initiation” was defined relative to the sampling date rather than symptom onset. This approach may have biased the assessment of antiviral effectiveness in some patients; therefore, this limitation should be considered when interpreting the results. Because imaging was obtained only when ordered by the treating clinician rather than systematically, estimates of radiological pneumonia may be subject to verification bias and should be interpreted with caution. Taken together, these strengths and limitations contextualize the results and support the overall robustness and interpretability of the observed associations.

## 5. Conclusions

In this post-pandemic cohort of hospitalized adults, in-hospital mortality was shaped predominantly by host-related vulnerability—most notably advanced age, substantial comorbidity burden, immunosuppression, and secondary bacterial infection—while viral etiology, including SARS-CoV-2, did not independently predict death after adjustment for these factors. These findings demonstrate that, in high-risk adult populations, baseline clinical fragility outweighs pathogen-specific differences in determining severe outcomes. Strengthening adult immunization programs, enhancing respiratory virus surveillance, preventing bacterial superinfection, and accelerating the development and equitable availability of effective vaccines and antiviral therapies remain essential strategies to reduce the burden of severe viral respiratory illness in hospitalized adults.

## Figures and Tables

**Figure 1 viruses-17-01545-f001:**
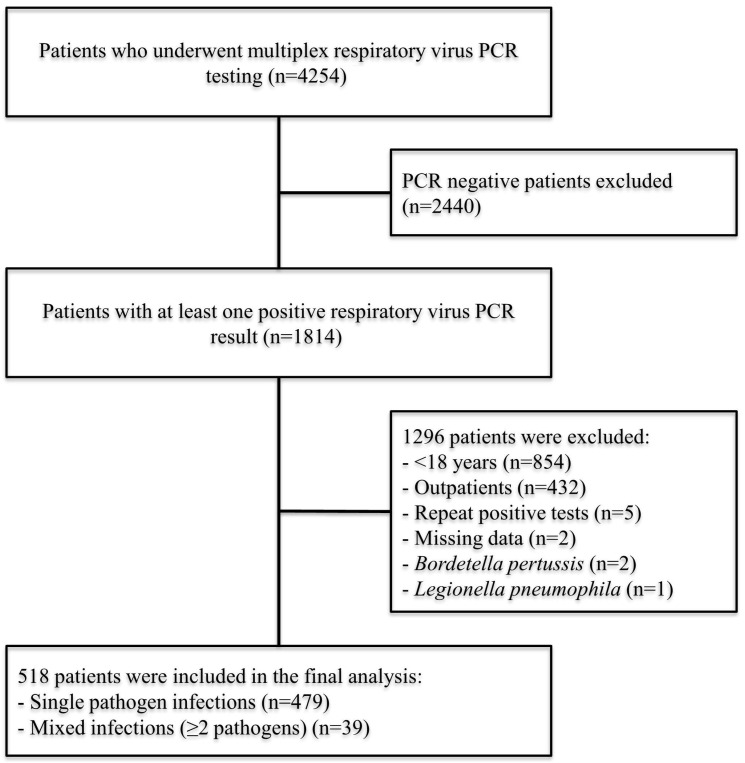
Flowchart of patient inclusion and exclusion process.

**Figure 2 viruses-17-01545-f002:**
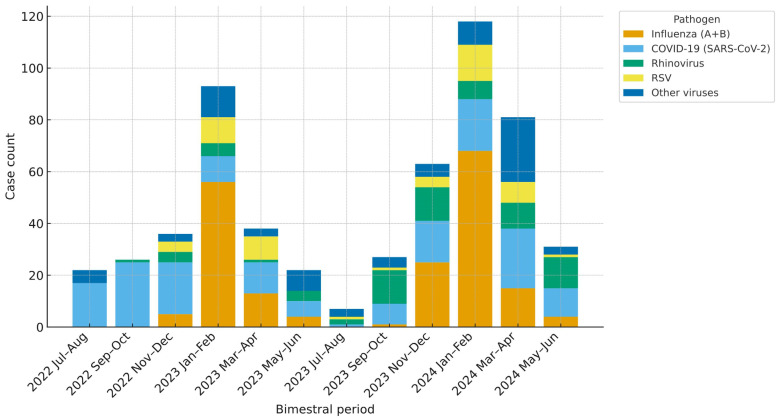
Pathogen Distribution by Bimonthly Period (July 2022–June 2024). Stacked bars show two-month counts in hospitalized adults. Influenza (A + B) pools A and B. Co-infections are counted per pathogen. “Other viruses” = adenovirus, parainfluenza 1–4, seasonal HCoVs (229E/NL63/OC43/HKU1), human metapneumovirus, enterovirus.

**Table 1 viruses-17-01545-t001:** Demographic and Clinical Characteristics of Patients.

	All Patients (n = 518)
**Age, mean ± SD [min–max]**	62.8 ± 17.6 [18–94]
**Sex** Male, n (%)	282 (54.4)
**Presence of comorbidity, n (%)** ^†^ Diabetes mellitus Cardiovascular disease Chronic lung disease (COPD, asthma) Hematological malignancy Solid tumor	476 (91.9) 145 (28.0) 143 (27.6) 104 (20.1) 102 (19.7) 98 (18.9)
**Charlson Comorbidity Index, median (IQR) [min–max]**	5 (2–7) [0–15]
**Immunosuppression, n (%)**	223 (43.1)
**Vaccination status, n (%) ^‡^** Influenza vaccination COVID-19 vaccination Pneumococcal Vaccination	37 (7.1) 469 (90.5) 151 (29.2)
**Clinical findings at the time of sample collection, n (%) ^§^** Fever Cough Dyspnea	161 (31.1) 279 (53.9) 161 (31.1)
**Oxygen requirement at sample collection, n (%)**	262 (50.5)
**Laboratory findings at the time of sample collection** Leukocyte count (cells/µL), median (IQR) Neutrophil count (cells/µL), median (IQR) Lymphocyte count (cells/µL), median (IQR) CRP (mg/L), median (IQR)	7500 (4600–11,350) 5540 (2937–8742) 870 (520–1400) 74.0 (33.0–166.0)
**Radiological evidence of pneumonia, n (%)**	109 (21.0)
**Secondary bacterial infection, n (%)**	53 (10.2)
**Location at the time of sample collection, n (%)** Ward ICU	361 (69.7) 157 (30.3)
**ICU admission after sample collection, (n = 361), n (%) ^¶^**	96/361 (26.6)
**Length of stay in ward (days) median (IQR) [min–max]**	10 (4–19) [0–143]
**Length of stay in ICU (days), median (IQR) [min–max]**	0 (0–10) [0–176]
**Total length of hospital stay (days), median (IQR) [min–max]**	16 (9–29) [0–176]
**Status on day 30, n (%)** Discharged Follow-up ongoing In intensive care unit Deceased	299 (57.7) 52 (10.0) 44 (8.5) 123 (23.7)
**Overall in-hospital mortality, n (%)**	163 (31.5)
**Time from sample collection to death, median (IQR) [min–max] (n = ** **163)**	14 (6–25) [0–173]
**Time from hospital admission to death, median (IQR) [min–max] (n = 163)**	23 (13–37) [2–176]

ICU, intensive care unit; IQR, interquartile range; SD, standard deviation. ^†^ Some patients had more than one chronic disease. ^‡^ Vaccination definitions are detailed in the Methods section. **^§^** Only the three most common findings are shown. ^¶^ Calculated among patients not in the ICU at the time of sample collection.

**Table 2 viruses-17-01545-t002:** Distribution of Respiratory Viruses Detected by Multiplex PCR.

Detected Pathogen	n	%
SARS-CoV-2	169	32.6
Influenza A	167	32.2
Rhinovirus	72	13.9
Respiratory syncytial virus (RSV) A + B	52	10.0
Influenza B	24	4.6
Parainfluenza virus 3	19	3.7
Human metapneumovirus	15	3.0
Human coronavirus OC43	11	2.1
Adenovirus	10	1.9
Human coronavirus 229E	6	1.2
Human coronavirus HKU1	6	1.2
Human coronavirus NL63	6	1.2
Enterovirus	3	0.6
Parainfluenza virus 2	2	0.4
Parainfluenza virus 1	1	0.2
Parainfluenza virus 4	1	0.2
Total detected pathogens	564	1.089 per sample
Total positive samples	518	100.0
Mixed infections (≥2 pathogens)	39	7.5

Counts are based on positive samples of unique patients. Mixed infections indicate ≥2 pathogens in the same sample.

**Table 3 viruses-17-01545-t003:** Comparison of Clinical and Laboratory Characteristics According to Detected Virus ^†^.

Variable	Influenza (A + B) (n = 174) ^‡^	SARS-CoV-2 (n = 152)	Rhinovirus (n = 58)	RSV (n = 34)	*p*-Value
Age, mean ± SD	64.2 ± 17.6	64.5 ± 16.4	60.9 ± 19.6	62.3 ± 13.1	0.488 *
Male sex, n (%)	86/174 (49.4)	79/152 (52.0)	34/58 (58.6)	24/34 (70.6)	0.117 **
Any comorbidity, n (%)	158/174 (90.8)	140/152 (92.1)	51/58 (87.9)	32/34 (94.1)	0.726 **
CCI, median [IQR]	4.5 [2.0–7.0]	5.0 [2.5–7.0]	4.0 [2.0–6.0]	4.5 [3.0–7.0]	0.455 *
Immunosuppression, n (%)	57/174 (32.8) ^a^	72/152 (47.4) ^b^	22/58 (37.9) ^ab^	22/34 (64.7) ^b^	**<0.001 ****
Vaccination status, n (%)					
- Influenza vaccination	11/174 (6.3)	12/152 (7.9)	3/58 (5.2)	4/34 (11.8)	0.630 **
- COVID-19 vaccination	157/174 (90.2) ^a^	141/152 (92.8) ^a^	47/58 (81.0) ^a^	33/34 (97.1) ^a^	**0.035 ****
**Clinical and laboratory characteristics at the time of sample collection**					
- Fever, n (%)	60/174 (34.5) ^a^	45/152 (29.6) ^a^	7/58 (12.1) ^b^	6/34 (17.6) ^ab^	**0.005 ****
- Cough, n (%)	97/174 (55.7)	73/152 (48.0)	32/58 (55.2)	20/34 (58.8)	0.456 **
- Dyspnea, n (%)	45/174 (25.9)	58/152 (38.2)	14/58 (24.1)	10/34 (29.4)	0.068 **
- Oxygen requirement, n (%)	84/174 (48.3)	86/152 (56.6)	27/58 (46.6)	20/34 (58.8)	0.312 **
- IN ICU, n (%)	51/174 (29.3)	60/152 (39.5)	17/58 (29.3)	9/34 (26.5)	0.174 **
- WBC (cells/µL), median [IQR]	7.4 [4.8–11.4] ^b^	7.2 [4.3–11.3] ^b^	9.9 [6.4–14.0]^a^	7.7 [0.9–10.5] ^b^	**0.010 ***
- ALC (cells/µL), median [IQR]	0.89 [0.55–1.35]	0.86 [0.52–1.36]	0.96 [0.55–1.98]	0.76 [0.31–1.41]	0.178 *
- CRP (mg/L), median [IQR]	63.0 [35.2–142.2]	78.0 [33.0–173.0]	98.0 [28.2–160.8]	67.0 [52.0–143.0]	0.588 *
- Radiological evidence of pneumonia, n (%) ^§^	26/174 (14.9)	23/152 (15.1)	17/58 (29.3)	6/34 (17.6)	0.070 **
Secondary bacterial infection, n (%)	9 (5.2)	20 (13.2)	7 (12.1)	2 (5.9)	0.062 **
ICU admission after sample collection, n (%)	26/174 (14.9) ^a^	39/152 (25.7) ^a^	6/58 (10.3) ^a^	4/34 (11.8) ^a^	**0.016 ****
Total length of hospital stay (days), median [IQR]	14.0 [9.0–26.0]	17.5 [9.0–28.5]	14.0 [8.0–23.0]	18.5 [10.5–28.5]	0.158 *
In-hospital mortality, n (%)	45/174 (25.9) ^ac^	62/152 (40.8) ^b^	8/58 (13.8) ^a^	13/34 (38.2) ^bc^	**<0.001 ****
Time from sample collection to death, (days), median [IQR] (n= 128)	9 [6–30]	14 [7–23]	17 [4–23]	14 [6–20]	0.959 *
Time from admission to death (days), median [IQR] (n = 128)	20 [11–37]	22 [13–35]	26 [16–32]	20 [16–42]	0.994 *

ICU, intensive care unit; CCI, Charlson Comorbidity Index; WBC, white blood cell count; ALC, absolute lymphocyte count; CRP, C-reactive protein; IQR, interquartile range; SD, standard deviation. Bold highlights significant difference, *p* < 0.05. * Kruskal–Wallis Test, ** Pearson Chi-Square Test. ^†^ Pathogen-specific columns include only monoinfections (co-detections excluded); other respiratory viruses (adenovirus, parainfluenza viruses, human metapneumovirus, and seasonal coronaviruses [229E, OC43, HKU1, NL63]) were excluded due to small numbers. ^‡^ Influenza A and B were combined due to the low number of Influenza B cases. ^§^ Radiological pneumonia indicates documented cases (imaging obtained at clinician discretion); percentages use total denominators; missing not imputed. Overall *p*-values are shown in the last column. Within each row, different superscript letters denote significant pairwise differences after Holm correction (*p* < 0.05); groups sharing the same letter do not differ.

**Table 4 viruses-17-01545-t004:** Clinical severity and antiviral therapy—Influenza vs. SARS-CoV-2 ^†^.

Variable	Influenza (A + B) (n = 174)	SARS-CoV-2 (n = 152)	*p*-Value
**Key laboratory marker, median [IQR] ^‡^**			
Absolute lymphocyte count (cells/µL)—Day 7	985 (730–1645)	820 (375–1500)	**0.012 ***
**Treatment characteristics**			
Antiviral therapy (oseltamivir-molnupiravir), n (%):	150 (86.2)	50 (32.9)	**<0.001 ****
Antiviral initiation ≤48 h, n (%):	146/150 (97.3)	47/50 (94.0)	1.000 **
**Clinical severity and mortality**			
**Respiratory support (highest level during stay), n (%)**			
- Oxygen requirement (any)	93 (53.4)	98 (64.5)	**0.044 ****
- Low-flow (nasal cannula + simple mask)	44 (25.3)	40 (26.3)	0.832 **
- High-flow/reservoir mask	8 (4.6)	15 (9.9)	0.064 **
- Mechanical ventilation (IMV or NIMV)	41 (23.6)	43 (28.3)	0.330 **
ICU admission, n (%)	72 (41.4)	86 (56.6)	**0.006 ****
In-hospital mortality (overall), n (%)	45 (25.9)	62 (40.8)	**0.004 ****
Death by day 7	7 (4.0)	8 (5.3)	0.594 **
Death by day 14	14 (8.0)	19 (12.5)	0.184 **
Death by day 28	28 (16.1)	43 (28.3)	**0.008 ****
Total length of hospital stay (days), median (IQR)	14.0 (9.0–26.0)	17.5 (9.0–28.5)	0.297 *

IMV, invasive mechanical ventilation; NIMV, non-invasive mechanical ventilation; ICU, intensive care unit; IQR, interquartile range. Bold highlights significant difference, *p* < 0.05. * Mann–Whitney U test, ** Pearson Chi-Square Test. ^†^ Pathogen-specific columns include only monoinfections; samples with ≥2 pathogens (co-detections) were excluded from this comparison. ^‡^ All other laboratory results (including WBC and CRP at Day 0 and Day 7, and baseline absolute lymphocyte count) are presented in Appendix A.

**Table 5 viruses-17-01545-t005:** Univariable Comparison: Survivors vs. Non-survivors.

Variable	All Patients (n = 518)	Survivors (n = 355)	Non-Survivors (n = 163)	*p*-Value
Male sex, n (%)	282/518 (54.4)	193/355 (54.4)	89/163 (54.6)	1.000 **
Age, years, median (IQR) [min–max]	66.0 (51.0–76.0) [18–94]	64.0 (46.5–74.0) [18–94]	70.0 (59.0–79.5) [24–93]	**<0.001 ***
CCI, median (IQR) [min–max]	5.0 (2.0–7.0) [0–15]	4.0 (2.0–6.0) [0–13]	6.0 (4.0–7.5) [0–15]	**<0.001 ***
Immunosuppression, n (%)	223/518 (43.1)	137/355 (38.6)	86/163 (52.8)	**0.003 ****
Oxygen requirement at sampling, n (%)	262/518 (50.6)	129/355 (36.3)	133/163 (81.6)	**<0.001 ****
In ICU at sampling, n (%)	157/518 (30.3)	60/355 (16.9)	97/163 (59.5)	**<0.001 ****
Virus group (monoinfections), n (%)				**0.026 ****
Influenza (A + B)	174 (36.3)	129 (74.1)	45 (25.9)	
SARS-CoV-2	152 (31.7)	90 (59.2)	62 (40.8)	
Rhinovirus	58 (12.1)	50 (86.2)	8 (13.8)	
RSV	34 (7.1)	21 (61.8)	13 (38.2)	
Other viruses ^†^	61 (12.7)	42 (68.9)	19 (31.1)	
Mixed infections (≥2 pathogens), n (%)	39/518 (7.5)	23/355 (59.0)	16/163 (41.0)	0.247 **
WBC (cells/µL) at sampling, median (IQR) [min–max]	7500 (4600–11,350) [0–125,910]	7200 (4500–10,405) [20–52,930]	8770 (5080–13,180) [0–125,910]	**0.017 ***
ALC (cells/µL) at sampling, median (IQR) [min–max]	870 (520–1400) [0–114,170]	960 (552–1510) [0–33,600]	690 (355–1190) [0–114,170]	**<0.001 ***
CRP (mg/L) at sampling, median (IQR) [min–max]	74.0 (33.0–166.0) [0.6–622.0]	61.0 (23.0–145.0) [0.6–213.0]	117.0 (54.4–197.0) [8.0–622.0]	**<0.001 ***
Radiological evidence of pneumonia, n (%) ^‡^	109/518 (21.0)	50/355 (14.1)	59/163 (36.2)	**<0.001 ****
Secondary bacterial infection, n (%)	53/518 (10.2)	18/355 (5.1)	35/163 (21.5)	**<0.001 ****
Mechanical ventilation (any), n (%)	111/518 (21.4)	22/355 (6.2)	89/163 (54.6)	**<0.001 ****
ICU admission after sampling (among ward at sampling), n (%) ^§^	96/361 (26.6)	40/295 (13.6)	56/66 (84.8)	**<0.001 ****
Total length of hospital stay, days, median (IQR) [min–max]	16.0 (9.0–29.0) [0–176]	13.0 (8.0–25.0) [0–158]	23.0 (13.0–37.0) [2–176]	**<0.001 ***

CCI, Charlson Comorbidity Index; WBC, white blood cell count; ALC, absolute lymphocyte count; CRP, C-reactive protein; ICU, intensive care unit; IQR, interquartile range. Bold highlights significant difference, *p* < 0.05. * Mann–Whitney U test, ** Pearson Chi-Square Test. ^†^ Other viruses include adenovirus, parainfluenza viruses, human metapneumovirus, and less common coronaviruses (229E, OC43, HKU1, NL63). ^‡^ Radiological pneumonia indicates documented cases (imaging obtained at clinician discretion); percentages use total denominators; missing not imputed. ^§^ Calculated among patients who were in the ward at the time of sampling.

**Table 6 viruses-17-01545-t006:** Multivariable Logistic Regression Analysis for Predictors of In-Hospital Mortality.

Variable	Odds Ratio (OR)	95% CI	*p*-Value
Age (per 10 years)	1.25	1.00–1.57	**0.048**
Sex (male)	0.82	0.49–1.39	0.467
Charlson Comorbidity Index (per point)	1.08	0.96–1.22	0.207
Immunosuppression	3.67	1.95–6.89	**<0.001**
Secondary bacterial infection	7.00	3.13–15.66	**<0.001**
ICU at sampling	5.52	3.00–10.16	**<0.001**
Oxygen requirement at sampling	3.39	1.86–6.18	**<0.001**
Virus group (ref = Influenza)			
SARS-CoV-2	1.74	0.95–3.18	0.073
Rhinovirus	0.29	0.11–0.78	**0.014**
RSV	2.01	0.75–5.37	0.165
Other viruses	1.56	0.66–3.69	0.308
White blood cell count (per 1000 cells/µL)	1.04	1.00–1.09	**0.036**
Absolute lymphocyte count (per 100 cells/µL)	0.99	0.99–1.00	**0.037**
CRP (per 10 mg/L)	1.01	0.99–1.03	0.269

OR, odds ratio; CI, confidence interval; ICU, intensive care unit; CRP, C-reactive protein. Bold indicates statistical significance at *p* < 0.05.

## Data Availability

The datasets used and/or analyzed during the current study are available from the corresponding author on reasonable request.

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
