# Peer review of "Comparative Clinical Outcomes of Major Respiratory Viruses in Hospitalized Adults During the Post-Pandemic Period: A Retrospective Cohort Study"

_viruses, 2025, doi:10.3390/v17121545_

Round 1

Reviewer 1 Report

Comments and Suggestions for Authors

I congratulate the authors for their study. I find this manuscript very interesting however I would advise for

1) More extensive discussion on respective studies of viral surveillance is pivotal e.g Hong S et al., 2025 – China, tertiary hospital, Pizzo M et al., 2025 – Italy, adults hospitalized with acute respiratory syndromes and several others please

2) No mention on complications / organ dysfuction eg while in ICU or respective care recorded, as well as presence of bacterial co-infections - all of the above are important for outcomes

Author Response

Response to Reviewer 1

  • More extensive discussion on respective studies of viral surveillance is pivotal e.g Hong S et al., 2025 – China, tertiary hospital, Pizzo M et al., 2025 – Italy, adults hospitalized with acute respiratory syndromes and several others please

Response: Thank you for this valuable suggestion. In accordance with the reviewer’s recommendation, we expanded the Discussion to include recently published viral surveillance studies from different geographic regions. We incorporated the suggested Italian and Chinese cohorts, which provide relevant post-pandemic circulation data in hospitalized adult populations. Following these additions, the paragraph was reorganized to ensure coherence and maintain logical flow.

Page 11, Lines 263–267 

  • No mention on complications / organ dysfuction eg while in ICU or respective care recorded, as well as presence of bacterial co-infections - all of the above are important for outcomes

Response: Thank you for this important and constructive comment. In line with your suggestion, we systematically incorporated secondary bacterial infections into the Methods, Results, Tables, and Discussion sections of the manuscript. Tables 1, 3, 5, and 6 have been updated accordingly. These revisions are summarized below:

Methods section:

Data Collection--- secondary bacterial infections

Page 5, Line 107

Definitions

We added a clear definition of secondary bacterial infection “clinically significant bacterial growth in blood or respiratory cultures obtained during hospitalization” and included this variable among the recorded clinical outcomes.

Page 6, Lines 127-128

Results section:

We added the prevalence of secondary bacterial infections to Table 1 (overall cohort characteristics).

Page 8, Lines 184-185

Pathogen-specific secondary infection frequencies were incorporated into Table 3 (virus group comparisons).

Pages 8-9, Lines 202-204

The survivor vs. non-survivor comparison in Table 5 was updated to include secondary bacterial infection rates.

Page 9, Lines 228-229

Secondary bacterial infection was included in the multivariable model in Table 6, where it emerged as one of the strongest independent predictors of in-hospital mortality.

Page 10, Lines 235-243

Discussion section:

We integrated a new interpretation detailing the clinical relevance of secondary bacterial infection, its distribution across viral etiologies, and its role as a major determinant of mortality.

We also contextualized our findings with relevant literature demonstrating the established impact of bacterial superinfection in viral respiratory illnesses.

These comprehensive revisions address the reviewer’s concern and strengthen the manuscript by providing a more complete understanding of complications and co-infections within the clinical course of respiratory viral infections.

Page 11, Lines 280-282

Page 13, Lines 329-331

Page 14, Lines 338-343

Page 14, Lines 346-348

Abstract

The Abstract has been revised to reflect the updated results and newly integrated findings.

Page 2, Lines 30-43

Conclusion section

The conclusion section has been reorganized and updated to reflect the newly integrated findings.

Page 15, Lines 370-378

Thank you for raising this important point. Our study was primarily designed to assess respiratory severity, as respiratory failure represents the predominant and most clinically relevant form of acute organ impairment among adults hospitalized with viral respiratory infections. Accordingly, key indicators such as oxygen requirement, level of respiratory support, and ICU status at the time of sampling were incorporated into the dataset and formed the main parameters of acute clinical deterioration captured in our analysis.

Additional organ-specific variables (such as vasopressor use) were available in the source data; however, these were not incorporated into the analytical framework in order to maintain alignment with the original study aims. We appreciate the reviewer’s attention to this aspect and acknowledge this as a relevant area for future work.

Reviewer 2 Report

Comments and Suggestions for Authors

The manuscript is well written, timely, and tackles an important question in the post-pandemic landscape: how major respiratory viruses compare in terms of severity and mortality among hospitalized adults. The study is solidly conducted, the statistical approach is appropriate, and the results are coherent with existing international evidence. I believe the manuscript is suitable for publication after a small number of clarifications that would enhance transparency and interpretability of the findings. 

a) Acknowledge more explicitly that the cohort represents a very high-risk population, with unusually high comorbidity burden and a large proportion of immunosuppressed patients, and explain briefly how this affects generalizability.

b) Provide context for the unexpectedly high overall mortality (31.5%), clarifying whether this is related to referral patterns, severity at presentation, ICU load, or the characteristics of the hospital.

c) Clarify that the timing of antiviral therapy was calculated relative to sampling rather than symptom onset, and explain how this limitation affects interpretation of early treatment, especially for influenza.

d) State more clearly in the Methods that the QIAstat-Dx panel reports rhinovirus and enterovirus as a combined target, and recall this in the Discussion so readers understand the heterogeneity of the “rhinovirus” group.

e) Add a brief note that radiological pneumonia was assessed only when imaging was ordered by the clinician, which may introduce verification bias.

f) Specify that no Omicron-adapted COVID-19 vaccines were available in Türkiye during the study period, and integrate this information when discussing the mortality observed in SARS-CoV-2 cases.

g) Comment briefly on the high mortality observed in RSV infections, relating it to the high prevalence of immunosuppression in that subgroup.

h) Add one or two sentences clarifying whether seasonal testing intensity or hospital policies may have influenced the apparent distribution of viruses over time.

Author Response

Response to Reviewer 2

  • Acknowledge more explicitly that the cohort represents a very high-risk population, with unusually high comorbidity burden and a large proportion of immunosuppressed patients, and explain briefly how this affects generalizability.

Response: We agree that our cohort reflects a particularly high-risk inpatient population characterized by advanced age, substantial comorbidity burden, and a high prevalence of immunosuppression. To improve clarity and interpretability, we have now incorporated an explicit statement in the Strengths and limitations section acknowledging these characteristics and noting that they may limit the generalizability of our findings to younger or healthier populations. This addition strengthens the transparency of the study and aligns with the reviewer’s recommendation.

Page 14, Lines 355-358

  • Provide context for the unexpectedly high overall mortality (31.5%), clarifying whether this is related to referral patterns, severity at presentation, ICU load, or the characteristics of the hospital.

Response: Thank you for this important and insightful comment. We agree that the overall mortality of 31.5% is high and warrants clarification. As our institution is a tertiary-care referral center, the cohort included a clinically complex population with substantial comorbidity burden — for example, 104 patients (20.1%) had hematological malignancies and 98 (18.9%) had solid organ malignancies, in addition to many other chronic conditions. Furthermore, a considerable proportion of patients were already in the ICU at the time of sampling or required subsequent ICU transfer, reflecting significant severity at presentation. These factors likely contributed to the elevated mortality observed. We have integrated this explanation into the Discussion to improve interpretability.

Page 13, Lines 318-322

Response:

  • Clarify that the timing of antiviral therapy was calculated relative to sampling rather than symptom onset, and explain how this limitation affects interpretation of early treatment, especially for influenza.

Response: As suggested, we now explicitly state in the Methods that antiviral timing was calculated relative to the sampling date because symptom-onset timing could not be reliably determined for all patients. We have also revised the Discussion to emphasize that this limitation affects the interpretation of early antiviral therapy—particularly for influenza, where treatment efficacy is highly time-dependent. The revised text now notes that, because treatment timing was defined relative to sampling rather than symptom onset, the measure reflects prompt in-hospital management rather than the true biological window in which early influenza therapy is most effective.

Page 5, Line 122

Page 12, Lines 303-307

Response:

  • State more clearly in the Methods that the QIAstat-Dx panel reports rhinovirus and enterovirus as a combined target, and recall this in the Discussion so readers understand the heterogeneity of the ‘rhinovirus’ group.

Response: Thank you for this suggestion. The Methods section already states that the QIAstat-Dx panel reports rhinovirus and enterovirus as a combined detection target. As recommended, we have now added a brief clarification in the Discussion to remind readers that the “rhinovirus” group represents a heterogeneous combined category and should be interpreted accordingly.

Page 12, Lines 296-298

  • Add a brief note that radiological pneumonia was assessed only when imaging was ordered by the clinician, which may introduce verification bias.

Response: In accordance with your suggestion, we have revised the wording in the Strengths and Limitations section to more clearly state that radiological pneumonia was assessed only when imaging was ordered by the treating clinician. The sentence now reads: “Because imaging was obtained only when ordered by the treating clinician rather than systematically, estimates of radiological pneumonia may be subject to verification bias and should be interpreted with caution.”

Page 15, Lines 364-366

  • Specify that no Omicron-adapted COVID-19 vaccines were available in Türkiye during the study period, and integrate this information when discussing the mortality observed in SARS-CoV-2 cases.

Response: We have now clarified in the Methods that Omicron-adapted COVID-19 vaccines were not available in Türkiye during the study period. We also highlight this point in the Discussion, where we note that Omicron-adapted vaccines were not introduced in Türkiye and none of the patients had received variant-specific boosters, which may partly explain the sustained mortality observed among SARS-CoV-2 cases despite prior immunization.

Page 5, Lines 114-115

Page 13, Lines 310-312

  • Comment briefly on the high mortality observed in RSV infections, relating it to the high prevalence of immunosuppression in that subgroup.

Response: As suggested, we have added a brief clarification in the Discussion noting that the relatively high mortality observed among RSV cases likely reflects the substantially higher prevalence of immunosuppression in this subgroup.

Page 11, Lines 276-277

  • Add one or two sentences clarifying whether seasonal testing intensity or hospital policies may have influenced the apparent distribution of viruses over time.

Response: As recommended, we clarified in the Methods that MRT-PCR testing during the study period followed stable, symptom-based criteria without changes in hospital policy. We also added a statement in the Discussion noting that the alignment of our findings with global surveillance patterns indicates that the observed temporal distributions likely reflect genuine epidemiological dynamics rather than artifacts of seasonal testing intensity. These revisions improve the interpretability of the time-trend analyses.

Page 6, Lines 139-142

Page 11, Lines 260-262

Reviewer 3 Report

Comments and Suggestions for Authors

The scientific article's topic is relevant, and the results presented in it will have independent significance and can be used as material for meta-analysis and writing systematic reviews. I would like to take a moment to share a few observations on the matter.

(1) Abstract. Please note that 19.6% of the cases were due to other, non-listed viral infections.

(2) The method used to determine the degree of immunosuppression is not clear.

(3) Of the 163 deaths, 40 patients died outside the intensive care unit. However, the authors did not disclose the causes of death in this case. It is noteworthy that the majority of deaths are not associated with severe viral pneumonia, as pneumonia was reported in only 109 (21.0%) patients, and hospital mortality was 163 (31.5%). Conversely, the majority of other studies have demonstrated a more explicit correlation between mortality from viral infections and severe pneumonia.

(4) It is evident that the authors did not provide any data concerning the SARS-CoV-2 variant.

(5) Table 1, it is unclear what the authors mean by fever: >37 oC or >38 oC (SIRS criterion).

(6) References should be formatted according to the MDPI style.

Author Response

Reviewer 3

  • Please note that 19.6% of the cases were due to other, non-listed viral infections.

Response: Thank you for this observation. We have revised the Abstract to state that other respiratory viruses accounted for 19.6% of all detections.

Page 2, Line 31

  • The method used to determine the degree of immunosuppression is not clear.

Response: We have clarified the definition of immunosuppression in the Methods section. Immunosuppression was defined as active cancer therapy or transplant; systemic corticosteroids ≥10 mg prednisolone-equivalent for ≥14 days; cytotoxic or biologic immunomodulators; primary immunodeficiency; or advanced HIV infection. Patients meeting any of these criteria were classified as immunosuppressed.

Page 5, Lines 117-120

3) Of the 163 deaths, 40 patients died outside the intensive care unit. However, the authors did not disclose the causes of death in this case. It is noteworthy that the majority of deaths are not associated with severe viral pneumonia, as pneumonia was reported in only 109 (21.0%) patients, and hospital mortality was 163 (31.5%). Conversely, the majority of other studies have demonstrated a more explicit correlation between mortality from viral infections and severe pneumonia.

Response: Thank you for this important and insightful comment. In accordance with your suggestion, we expanded the Discussion to clarify the clinical context of deaths in our cohort. As now detailed in the revised manuscript, our study population reflects the high clinical complexity of a tertiary-care referral center, with non-survivors being older, more comorbid, more often immunosuppressed, and frequently requiring ICU-level care at or shortly after sampling. In this vulnerable population, viral infection may precipitate terminal decompensation of advanced malignancy, heart failure, or other end-stage conditions, leading to mortality even in the absence of severe viral pneumonia. This helps explain why the overall mortality (31.5%) exceeded the proportion of patients with radiologically confirmed pneumonia (21.0%), in contrast to patterns described in many other studies. We also note that imaging was obtained selectively rather than systematically, which may have resulted in under-recognition of lower respiratory tract involvement—a limitation inherent to the retrospective design.

Page 13, Lines 325-329

As our study was designed to assess all-cause in-hospital mortality rather than to adjudicate specific causes of death, the immediate cause of death was not systematically recorded. Consequently, cause-specific mortality could not be reliably analyzed within the scope of this retrospective dataset. These clarifications have now been incorporated into the revised manuscript to enhance transparency and interpretability of the mortality findings.

4) It is evident that the authors did not provide any data concerning the SARS-CoV-2 variant.

Response: Variant-level data were not available in our dataset, as routine genomic sequencing was not performed during the study period. We now clarify this point in the Methods to avoid misinterpretation.

Page 6, Line 145

5) Table 1, it is unclear what the authors mean by fever: >37 oC or >38 oC (SIRS criterion).

Response: In the revised manuscript, we clarified in the Methods section that fever was defined as a measured temperature ≥38.0°C, consistent with standard clinical criteria. Accordingly, Table 1 should now be interpreted using this definition.

Page 5, Line 120

6) References should be formatted according to the MDPI style.

Response: All references have now been thoroughly revised and reformatted according to the official MDPI reference style.